# Pointing Tasks using Spatial Audio on Smartphones for People with Vision Impairments

Category: Research

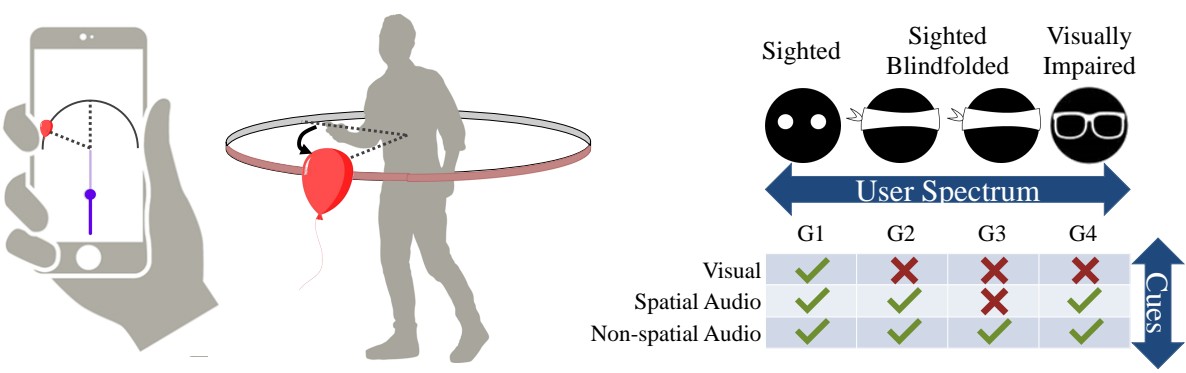

(a) Interactive context: Mobile shooting game      (b) Experiment Strategy: Sequential cue deprivation

Figure 1: Our interactive context is defined by (a) a simple shooting game that captures the core interaction task of orientation in a typical AR application on a smartphone (b) sequential cue reduction to understand the effect of removing visual and spatial auditory cues in a sequential manner to comparatively study their effects with respect to PVIs where the visual cue is absent. We define four user groups: G1, G2, G3, and G4 based on the cues provided.

## ABSTRACT

We present an experimental investigation of spatial audio feedback using smartphones to support direction localization in pointing tasks for people with visual impairments (PVIs). We do this using a mobile game based on a bow-and-arrow metaphor. Our game provides a combination of spatial and non-spatial (sound beacon) audio to help the user locate the direction of the target. Our experiments with sighted, sighted-blindfolded, and visually impaired users shows that (a) the efficacy of spatial audio is relatively higher for PVIs than for blindfolded sighted users during the initial reaction time for direction localization, (b) the general behavior between PVIs and blind-folded individuals is statistically similar, and (c) the lack of spatial audio significantly reduces the localization performance even in sighted blind-folded users. Based on our findings, we discuss the system and interaction design implications for making future mobile-based spatial interactions accessible to PVIs.

**Index Terms:** Human-centered computing—Auditory Feedback; Computing methodologies—Mixed / Augmented reality; Human-centered computing—Empirical studies in accessibility

## 1 INTRODUCTION & BACKGROUND

The motivation for our research is to enable people with vision impairments (PVIs) to experience and use augmented reality (AR) applications without requiring expensive investment in hardware. Specifically, our quest is to enable PVIs to perform pointing tasks in 3D in smartphone-based AR applications. Today, many AR applications ranging from games and entertainment to education and training run on commodity smartphones, optionally with very low cost headsets. For instance, *Pokemon Go* [11], one of the fastest mobile AR application to reach US$ 100 million in sales, runs on smartphones with no required add-ons. Further, the capabilities of smartphones are rapidly increasing along multiple dimensions: increased processing power, including built in GPUs, multiple high resolution cameras, a host of sensors, including GPS, IMU, Lidar and others, and 5G connectivity that brings the power of cloud AI/ML within the reach of most smartphone users. Thus powerful AR applications that harness these capabilities are beginning to

be available in diverse domains [19, 54, 73, 88]. Riding on the same capabilities of smartphones, a newer generation of virtual reality applications, called Lightweight VR [17] that provide semi-immersive VR experiences on smartphones are beginning to appear. However, most of the efforts previously made to make VR accessible for PVIs proposed systems using expensive, bulky, or custom-built hardware including hapic devices and gloves, head mounted devices, custom earpieces etc. which are highly application oriented [26, 51, 52, 68]. Use of such complex hardware challenges the scalability of these proposed techniques for accessible AR/VR. The increasing versatility of modern day smartphones can be utilized to tackle the hardware challenges to some extent [23].

Despite these advances, very little work has been done in creating tools that enable the above experiences to be available to PVIs (see Section 1.2). In this paper, we present the very early steps that we have taken to address this gap. Given the paucity of such work, our goal is to investigate hand-held AR or VR, where the smartphone is held by the user in their hand and moved around to experience the application. Thus we are not considering head-worn AR or VR, which will present more complexities by requiring an additional device to do the pointing or selecting of direction.

We limit our focus to the operation that is fundamental to all such handheld AR/VR applications, that of pointing at an object in 3D space or orienting towards a direction in 3D space, usually by moving the smartphone around. This task is usually accomplished by the sighted users by locating the object of interest visually as they pan the phone around to search for the object. The basic question we seek to address is, how does a person without sight accomplish the same task? More specifically, how can spatial audio be used to accomplish this task in the AR/VR environment?

While there is an extensive volume of work [18, 25, 31, 42] on how PVIs perceive through other senses such as sound or touch, how these senses can be meaningfully integrated into a usable interactive AR experience on a smartphone requires deeper inquiry of sensory effects within concrete interactive contexts.

## 1.1 Problem & Research Questions

In this paper, the question we seek to address is: Can users restricted to using just the auditory sense achieve task performance similar to the one achieved by users with vision? We narrow our focus to the specific task of *orientation* i.e., the ability to localize the direction of an object from the egocentric frame of reference of an individual [47].

Auditory feedback for orientation has been has been extensively studied in perceptual psychology [45, 49, 55, 85]. Zahorik et al. [85] report that binaural cues (cues independently transmitted to each ear separately) play a critical role in direction localization. On the other hand, distance perception is affected by a wide range of cues including intensity and spectrum. At the very least, it is understood that combining audio cues effectively for an operational AR environment requires a systematic and controlled series of studies. In this paper, we begin by asking the following questions, in the context of AR with smartphones:

1. What is the difference, if any, between the performance of blindfolded sighted users and PVIs for orientation tasks?

2. What are the similarities and dissimilarities between the actions and motor strategies of blindfolded sighted users and PVIs for execution of the task?

3. How much does spatial audio specifically affect the **accuracy of orientation** (pointing in a desired direction) in the absence of visual feedback (either through blindfolding or because of visual impairment)

4. How much does spatial audio specifically affect the **ability to sustain a given orientation** in the absence of visual feedback (either through blindfolding or because of visual impairment)

5. How much does spatial audio specifically affect **motor strategies** in enabling orientation in the absence of visual feedback (either through blindfolding or because of visual impairment)?

Here, by performance we mean the accuracy of and the time taken for locating an object placed around the individual's body. By actions and motor strategies We believe that a careful investigation of these questions will offer critical insights needed for integrating spatial audio feedback in a practicable manner for PVIs.

## 1.2 Prior Work

The exploration of multi-modal interfaces in virtual reality environments has a rich history starting way back in the sixties with Heilig's *sensorama* [41]. There are very many works that make fundamental contributions to the perception of spatial audio and its practical use in real and virtual environments. There are numerous past and ongoing research in the use of spatial audio in AR and VR environments (for example, see the special issue on spatial audio in VR, AR and MR [33] and articles therein, including the relatively newer area of 360 degree video [20]. Similarly there are many that examine the use of touch in VR systems [28, 56]. The benefits of combining spatial audio and haptics to enhance presence in such environments has been well recognized and there are many works exploring this combination of sensory inputs [29], but most of them are about enhancing the overall perception of presence, in conjunction with the visual display.

There are few works that discuss VR accessibility for PVIs. Maidenbaum et al. proposed a virtual *EyeCane* for navigation tasks in VR [57]. *PowerUp* laid out some guidelines for making web-based games accessible to PVIs [77]. *NavStick* utilizes a gaming controller to navigate their surroundings with speech as the primary sensory cue [63]. *SeeingVR* provide a set of tools such as text-to-speech, and magnification lens to aid people with low vision to see in VR [87].

The study by Dong et al. showed that in a VR environment using 3D auditory feedback, the experience of PVIs is different when compared with that of sighted people [35]. Some games developed in the past such as *VI-Tennis* and *AudiOdyssey* use a variety of hardware devices to provide a gaming experience for sighted and PVI gamers [40, 61]. Drossos et al. developed a computer based tic-tac-toe game with binaural sound effects for blind children [36]. A few games such as *VBGhost* and *TapBeats* utilize smartphones to play games accessible to PVIs [46, 59]. There are a variety of works which target learning for PVIs using audio-gamification approach [21, 53, 72]. This audio-gamification can also be utilized in various applications in AR/VR environments by making use of the capabilities of widely used smartphones.

In parallel, there have been explorations of utilizing spatial audio interfaces for providing directional and distance cues. One of the early works that inspires our approach is by Sanchez et al. [74], who demonstrated a game environment, *AudioDoom* to enable spatial learning for blind children. Frauenberger and Noisternig [38] later proposed a formal software implementation framework dubbed VAR (virtual audio reality) for smooth integration of auditory cues within VR systems. Kolsover et al. [50] integrated the primary manipulative senses, namely, visual, auditory, and haptic senses for providing directional cues in mobile navigation. A more recent work by Brill et al explores a combination of vibrotactile and spatial audio directional cues for pararescue jumpers in the U.S. Air Force [24]. There are a bunch of commercially available *Audio Games* which use a variety of auditory cues such as 3D sound [5–7, 16], stereo sound [1, 10, 14, 15] and, verbal cues [8, 12, 13]. *Audio Game Hub* is an iOS app available on App Store which also contains a bunch of audio games accessible to PVIs [2–4]. The use of spatial audio for enabling gamers without sight to play mainstream video games has been proposed in [76], but the work primarily deals with desktop or console gaming scenarios and has not been studied on smartphones.

There are many works that use audio, haptics or a combination of these to help PVIs to navigate in the real world. [34, 50, 64, 75, 78, 81]. Microsoft Soundscape [9] is an iOS app that uses the Geowiki Open Street Map to enable PVIs to navigate the real world using spatial audio to speak out points of interest. The use of spatial audio helps in the orientation and localization of points of interest. In addition, in a specific navigation mode, there is an audio beacon (a virtual drumbeat that is located at the coordinate of the destination) played out in spatialized audio that enables the user to determine the relative orientation to the destination. Ross et al. [70] address the broad questions around audio based explorations in virtual environments by extending Microsoft Soundscape. One of the key results from their study is that participants found estimation of the distance to be very challenging and required additional cues of known physical locations or physically being present in known locations. The work by Zhao et al., [86] combines audio with haptics to enable PVIs to navigate in a VR environment to give a better sense of presence by providing the equivalent of a white cane in VR, facilitated by a physical device that is worn by the user.

The visual experiences of a sighted individual play an important role in their development of spatial knowledge [66]. There are many works comparing sighted, sighted-blindfolded, and PVIs across a variety of studies across different age groups. Campus et al. compared sighted-blindfolded and early-blind subjects across spatial bisection, and temporal bisection tasks [30]. Ribadi et al. compared static and dynamic balance among these three groups in adolocents [67]. It was observed that sighted children show consistent improvement in spatial navigation skills with age as compared to blind children [37, 62]. An earlier study by Klatzky et al. found a significant difference across these groups in tasks involving spatial inference but didn't find a significant spatial deficit among PVIs [48]. Haptic material perception has also been studied across similar group of participants and no significant advantage has been recorded because of the visual

experiences [22]. Similarly, Rovira et al. didn't find any significant deficit among blind and sighted adolescents while performing mental rotation of 2D shapes [71]. Cattaneo et al. found differences in spatial bias between sighted individuals and PVIs, where PVIs exhibiting no significant spatial bias in vertical and radial dimensions [32]. Accessible games such as *BlindHero* also compared the performance across these groups [84]. Similar performance in spatial orientation and obstacle avoidance task using spatial audio was observed among sighted-blindfolded and PVI individuals in a study by Bujacz et al. [27].

### 1.3 Knowledge Gaps & Our work

We observe that a large portion of the body of work available on enabling both sighted as well as blind users deals with navigation tasks wherein the user is in motion. On the other hand, works that seek to support visually impaired users, while seminal, are largely application-oriented. Our work seeks to complement these works by offering a deeper task-oriented analysis of how and why spatial audio can enable orientation for PVIs in mobile AR. Secondly, our work establishes a crucial connection between the similarities and differences between sighted and visually impaired individuals. This connection serves as an important step in the development of design guidelines for sighted HCI designers who wish to incorporate the experience of PVIs in AR/VR environments while using sound and touch as the primary perceptual cues [35,51]. Finally, our work looks at the fine-grained process of direction localization in the absence of visual feedback. We specifically show that (a) the efficacy of spatial audio is relatively higher for PVIs than for blindfolded sighted users during the initial reaction time for direction localization, (b) the general behavior between PVIs and blind-folded individuals is statistically similar, and (c) the lack of spatial audio significantly reduces the localization performance even in sighted blind-folded users.

## 2 APPROACH & RATIONALE

There are three key aspects to our approach: (1) interactive context for a systematic study (Figure 1(a)), (2) perceptual cues for orientation, and (3) experimental strategy (Figure 1(b)).

### 2.1 Interactive Context

A simple application that captures this problem is the task of shooting a target at any orientation and distance with the user at the origin. Auditory localization depends on interaural loudness difference(IDL) and interaural time difference(ITD) and humans can take advantage of both during horizontal auditory localization tasks [60]. Vertical auditory localization is difficult which depends on the spectrum of sound cues as created by the outer ear, suggesting that vertical localization is difficult when compared with horizontal localization [58]. Hence, to simplify the study task, We restrict the target to be in the horizontal plane in our study. The basic task is for the users to orient the phone to align with the position of a target in the horizontal plane by moving the hand holding the phone around or by turning their body in place while holding the hand steady or a combination of the above. This is the core operation in most AR or lightweight VR applications and our goal is to make this task accessible to PVIs. The one change we make to a typical AR application is that instead of holding the phone perpendicular to the ground (in landscape or portrait mode) the users hold the phone with its surface parallel to the ground and use their other hand to interact with the touch screen.

And this can be used in other settings like wayfinding [39,69] or understanding the location of objects around the person in an augmented or mixed reality environment. Utilizing the metaphor of a *virtual cross-bow* [1], we designed and implemented a simple Android game app that generates a series of targets at different orientations and distances from the user. The user's task is to shoot the balloons using a slider in the smartphone screen akin to pulling

the string of a bow and releasing to shoot an arrow (Figure 1(a)). Effectively, the process of determining the shooting direction maps to the orientation task. Subsequently, the pulling of the slider to shoot the balloon maps to the user's ability to sustain a given orientation while focusing on a non-orientation task (i.e. hitting the balloon). Note that the estimation of distance of the balloon is not considered in our study.

### 2.2 Perceptual Cues

Given the context of our shooting game, we developed our game to include visual, auditory, and vibro-tactile feedback. Specifically, orienting the phone closer to the direction of balloon brings it within the visual range of the phone's screen. Furthermore, as the angular difference between phone and the balloon's direction reaches within a certain threshold, we also display a change in color of the balloon. Given that our primary target audience is PVIs, the reason for adding visual feedback was to establish a reference with respect to which we could understand how the absence of visual feedback would affect the performance and the actions of a user during the orientation task.

Our second and critical perceptual feedback for the orientation task is auditory. Here, we utilize spatial audio, which is a powerful evolutionary human ability that plays a role in drawing the visual sense roughly in the direction of the source of sound and then the visual sense accurately locates the source. Spatial audio as a powerful tool for people with vision impairment has long been recognized and there are many efforts that seek to exploit this sense for orientation [79,83]. On the other hand, Voss et al. show evidence for superior spatial hearing for blind individuals in the horizontal plane in addition to significant deficits in the vertical plane. [80]. Therefore, we specifically constrained the target balloons to be on the horizontal (azimuth) plane of the user which is further aligned with the display screen of the phone. In addition to spatial audio, we also added a non-spatial audio chirp feedback. These refer to beep tones with varying frequency of beeps to indicate angular deviation from the target. Such cues are quite common in both digital and physical environments (e.g. in accessible pedestrian crossings).

Finally, the association of the extent of the slider draw (pulling the string of the bow) is mapped to the vibrotactile feedback and its mapping to the distance of the target is an interesting area of future research. In this work, we "*cheat*" by providing a distinct "*lock*" sound when the target is in the correct range corresponding to the extent of the draw, prompting the user to release the virtual arrow. We then provide a verbal confirmation of the hit or the miss. The reason for this addition lock sound comes from extensive literature that essentially concludes enabling a person to accurately estimate the distance of an auditory source without visual feedback is currently prohibitively challenging if not completely impossible. Neilsen et al. [65] summarize the results of several auditory experiments in different room conditions and conclude, *"In the anechoic room there is no correspondence between physical and perceived distance"*.

### 2.3 Experimental Strategy

In order to systematically study the tasks of orientation and ranging, we implement a *sequential cue deprivation* strategy (Figure 1(b)). What we mean by that is we conducted a between-subjects experiment across four groups of users beginning with sighted users who played the game with all feedback mechanisms. This is our reference user group. Following this, we sequentially remove one cue at a time starting from vision, followed by spatial audio. This results in two groups, namely, sighted blind-folded users with spatial audio and beep cues and sighted blind-folded users with beep cues but without spatial audio. Our final group is comprised of the PVI users who are provided with both spatial audio as well as beep cues. These four groups allowed us to explore our research questions.

## 3  EXPERIMENTAL SETUP

We conducted controlled user studies where participants performed tasks involving spatial orientation of objects which required spatial cognition. We introduced these tasks to the users in a sequential fashion starting with orientation task, followed by a task requiring both orientation and ranging. These tasks were presented in the form of a 2D shooting game where the goal was to shoot a virtual balloon placed at a distance. Each task corresponds to a specific setting of the game. We *gamified* the study primarily to get better engagement as the participants perform tasks with increased difficulty levels [21].

### 3.1  Game Design

For this experiment, we developed an Android game in which the goal is to shoot a virtual balloon which is emitting a beep sound in 3D space. Unlike the visual information which can be processed in parallel, non-visual information is described as sequential or serial processing [43]. Hence, we provide one target (balloon) at a time. In this game, a balloon appears at different orientations and distances from the user. The targets are restricted to be on a horizontal plane in front of the user. In order to shoot, the user needs to align the phone in the direction of the balloon by moving their hand/arm with which they are holding the phone in the horizontal plane in front of them. The game provides Auditory, Visual, and Vibrotactile feedback to the users in order to help the user shoot the virtual balloon. Each feedback mode is described below.

- **Auditory Feedback:** In some experimental conditions, the beep sound is spatialized which means that the user is able to identify the beep sound direction using which the user may align the phone in direction of the beep sound. To help with the alignment, another auditory cue is provided in terms of the varying beep frequency of the balloon. The beep frequency is inversely proportional with the angular difference between the phone and the balloon in the horizontal plane. As the user aligns the phone towards the balloon, the beep frequency keeps on increasing and reaches a maximum when the phone is perfectly aligned with the balloon. The game also provides a distinct 'lock' sound whenever the amount of slider pulled correctly corresponds to the distance of the balloon from the user within a margin of $\pm 5$ units, given that the phone is pointing directly towards the balloon within an angular margin of $\pm 5$ degrees.

- **Vibrotactile Feedback:** The user also needs to estimate the distance of the balloon using a vertical slider on the smartphone. The main purpose of this feedback is to add complexity to the tasks rather than providing a means to do distance estimation. The game provides a vibrotactile feedback to the user whenever they pull the slider. This slider represents a virtual bow which the user is pulling by sliding their finger vertically downwards on the screen. The amplitude of the vibration is proportional to the amount by which the slider is pulled. The amplitude keeps on increasing until the slider value correctly corresponds to the distance between the user and the balloon. Upon further pulling of the slider, amplitude will remain constant.

- **Visual Feedback:** The game provides visual feedback in the form of the color of the balloon. The balloon color changes based on how the user is aligned with respect to the balloon and how much the slider is pulled back. Red color represents that the user is neither within the allowable angular threshold ($\pm 5$ degrees), nor the slider is pulled back by the correct amount. Yellow color represents that the alignment of the phone is correct i.e., the angular difference between the phone and the balloon is less than $\pm 5$ degrees but the slider value is incorrect. Blue color represents that the slider is pulled back by the correct amount i.e., within $\pm 5$ units of the correct value but

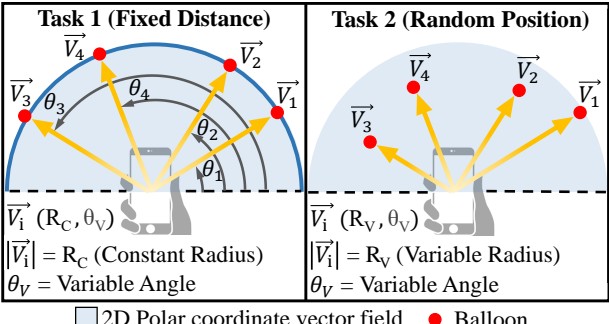

Figure 2: Users perform two tasks in the study. In Task 1 (left), the balloon is at a fixed distance ($R_C$) from the user with varying direction ($\theta_V$). In Task 2 (right), both distance and direction changes. $\vec{V_i}$ represents different trials.

the alignment error between the phone and the balloon is more than the threshold. Green color represents that the phone is aligned with the balloon direction and the slider value is also correct.

Once the users think that they are aligned with the balloon and the estimated distance is correct, they can shoot by lifting their finger from the screen. The game then gives a verbal confirmation of a hit or a miss. The game can be played in two modes – Practice mode and Game mode. In Practice mode, if the user misses the target, the target position will remain the same with respect to the user so that the user can try to shoot the same target again. In Practice mode, the target position will change only when the user successfully hits the target. In Game mode, the target position will change after every attempted shot irrespective of a hit or a miss i.e., the user only gets one chance to shoot the target.

## 4  EXPERIMENT DESIGN

Our experiment design was based on systematic approach to investigate how different sensory inputs help a person to locate an audio source in 3D space. We conducted controlled between-subjects experiments with four different group of participants. Each group performed the same experiment but under different physical conditions and sensory inputs. In the following sections we describe the participation pool, study tasks, experimental procedure, and evaluation metrics followed across all experiments.

### 4.1  Participants

We recruited a mix of 48 participants. Out of these, 36 were sighted participants and 12 participants had complete loss of vision. 3 sighted participants had prior experience with spatial audio through gaming consoles and prior user studies. We asked all the participants to wear headphones/earphones while playing the game in order to experience the spatial auditory cues. Other than the visual impairment, all the participants could comfortably use both of their upper limbs and had normal hearing capability.

We divided the 36 sighted participants into 3 user groups with 12 participants in each group. All the PVI participants formed the fourth group. The groups were divided based on physical condition of the participants and the sensory inputs provided while performing the experiment. The details about the different groups are as follows.

- **Group 1 (G1):** All the participants in this group played the game with their eyes open. Also, the beep sound of the balloon was spatialized i.e., the beep volume in the earphones would be different in right ear when compared to that in the left ear if the balloon is not exactly in front of the user.

- **Group** 2 (**G**2)**:** All the participants in this group played the game with their eyes closed. The beep sound of the balloon was spatialized similar to that of G1.

- **Group** 3 (**G**3)**:** Similar to G2, all the participants in group-3 played the game with their eyes closed with the difference that the beep sound of the balloon was not spatialized. Hence, the beep volume in both the ears was the same.

- **Group** 4 (**G**4)**:** All the Blind and Visually Impaired participants were grouped together. Similar to G1 and G2, the beep sound of the balloon was spatialized. G2 closely matches this group in terms of the physical conditions and sensory inputs.

## 4.2   Tasks

In this experiment, each user performed two tasks with a common goal of shooting the balloon. The tasks were separated based on how the balloon position changes with respect to the user across different trials. Below, we define the two tasks.

- **Task** 1 **– Fixed Distance:** In this task, the distance of the balloon with respect to the user is fixed and only the direction of the balloon changes between trials (Fig. 2 (left)). This means that the user will need to pull the slider the exact same amount in each trial to successfully hit the balloon. Between two consecutive trials, only the alignment of the balloon will change in the horizontal plane in front of the user.

- **Task** 2 **– Random Position:** In this task, both the distance and direction of the balloon changes between two consecutive trials (Fig. 2 (right)). The amount of slider pull required between two consecutive trials is different.

## 4.3   Procedure

Each study took approximately 45 minutes for the sighted individuals and around 1.5 hours with the PVIs. The study with sighted participants was done in person using the same phone with the app loaded while the study with PVIs had to be done online (since the pandemic induced restrictions on travel and meetings were in place where the study was done). Hence the study with PVIs took longer since all the instructions were given over phone while the participants were using the same device to download, install, trial and perform the tasks while the researcher continued to be on the call. The other differences due to this remote study are presented in the discussions section.

Each session started with a general introduction of the game to familiarize the user with the interface and different interaction techniques used in the game. Finally, they were asked to fill a demographic questionnaire before beginning with the study tasks.

Each of the sighted users shot the target 5 times in the Practice Mode of the game at the beginning of each task (Fixed Distance and Random Position) followed by the 5 study shots for each task in the Game Mode. In total, each user performed 10 practice shots and 10 study shots. The PVIs required upto 8 trial shots to understand the interface and to learn its use. They did the same 5 shots in the study phase for each task as the sighted participants. The practice shots helped the users to acquire adequate practice of the tasks before beginning with the study shots. All the users performed the tasks in a sequential manner. The users started the game by first performing the Fixed Distance task followed by the Random Position task. Participants in G2 and G3 were asked to close their eyes at all times while playing the game. Each trial took $15-30$ seconds to complete across all the user groups. In total, 960 trials were recorded across all the four user groups.

## 4.4   Data and Evaluation Metrics

For each trial performed by the participant, we recorded the raw event log containing (a) time taken for each trial, (b) phone's accelerometer data, (c) phone's gyroscope data, (d) balloon (target) position in 3D space, (e) no. of hits and misses, (f) position of the user in 3D space, (g) position of touch on the screen, (h) amount of slider pull, and (i) angular difference between the phone and the target. In order to analyze the actions performed by the user to achieve the goal of shooting the target, we take a closer look at the IMU data (rotation of the phone) and touch inputs (slider pulling) by the user. We further elaborate on this action analysis in the following sub-section.

### 4.4.1   Action Analysis

The users performed two actions in order to shoot the target i.e., rotate the phone and pull the slider. To analyze these actions, we plotted angular error between the target and the user v/s time, along with the slider value v/s time (Fig. 3 (a)) for every trial and observed a common trend in the motor strategy of the users across all the groups (Fig. 3 (b)). Based on our observations, we segmented these actions into four phases – Initial reaction time, Approach phase, Stabilization phase, and Slider pulling time. Each of these phases are defined below.

- **Initial Reaction Phase:** This phase starts when the trial starts i.e., the participant starts to hear the beeping sound. The phase ends when the user starts to move the phone either towards or away from the target, thus capturing the time taken by the user to react to the cues.

- **Approach Phase:** During this phase, the user moves the phone to align with the target. This phase follows immediately after the Initial reaction time ends and the user starts to show some movement of the phone. This phase ends when the angular error between the phone and the target first reaches zero or it reaches the minimum value recorded during that trial. This phase can be interpreted as the initial gross movement made by the user towards the target.

- **Stabilization Phase:** This phase represents the time spent by the user to do fine adjustments to the orientation of the phone in order to properly align with the target. In this phase we generally see a wavy pattern in the angular error v/s time plot and with every wave the angular error decreases, showing that the users generally took an iterative approach to do fine adjustments to the alignment. The phase starts immediately after the end of approach phase and ends as soon as the phone movement stops or becomes relatively small.

- **Slider Pulling Phase:** In this phase the user pulls the slider back to estimate the distance of the target. The phase starts when the user starts to pull the slider and ends when they stop pulling and release the slider by lifting their finger off the screen of the phone. This phase is independent of the other three phases, and it sometimes overlaps with the stabilization phase.

## 5   RESULTS

In the following sections, we report the statistical analysis of the user performance metrics. Furthermore we discuss the key findings and insights gained from our data collection, observation, and user-feedback from all trials performed by the participant. We present the analysis of the total time taken per trial for each user group across the two tasks. Subsequently, we segment each trial into different phases based on the general strategy used by the participants to shoot the target.

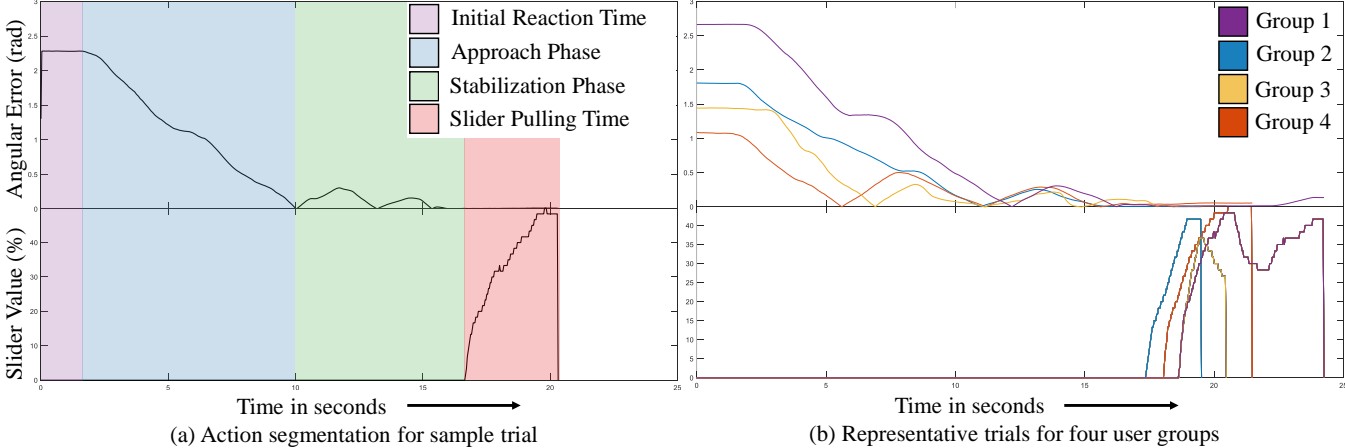

Figure 3: (a) We segment the actions of the users into four phases: Initial reaction time, Approach phase, Stabilization phase, and Slider pulling time. (b) A representative sample trial for each group, showing similarities in the actions and motor strategies across all groups.

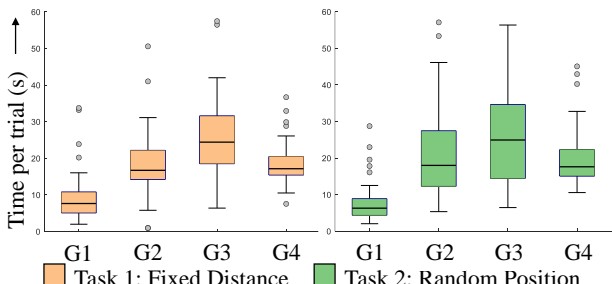

Figure 4: Time per trial for the four groups across the two tasks

## 5.1 Time Taken Per Trial

In this sub-section, we compare the total time taken by each participant per trial. We do this comparison across the two tasks for all user groups (See Fig. 4). We first tested the data for normality using Shapiro-Wilk test and found out that the data is not from a normal distribution. We further conducted hypothesis testing using Kruskal-Wallis test ($\alpha = 0.05$) which is the non-parametric statistical equivalent of one-way ANOVA test.

### 5.1.1 Comparison across tasks

We performed pair-wise comparison of the time taken by the participants in each group across the tasks. For this comparison, we considered group 1 as the gold standard because it had access to all the sensory cues.

For Task 1: Significant differences were observed between G2 ($p < 0.001$), G3 ($p < 0.001$), and G4 ($p < 0.001$) when compared to G1. This is along expected lines since G1 has the visual input while the others don't. Significant differences were observed between the mean time taken to complete each trial between G2 and G3 ($p = 0.01$), and G3 and G4 ($p = 0.03$). This brings out the contribution of spatial audio to the task at hand since G3 is deprived of spatial audio cues compared to G2 and hence takes longer to complete the task. What is significant is that sighted blindfolded users without the spatial audio cues took a longer time (M = 22.32 sec) compared to PVIs clearly establishing the value of spatial audio cues for orientation. It is important to note that no significant differences were found between G2 and G4 indicating that blindfolded sighted users and PVIs had similar task completion times for this task, though the higher standard deviation for G2 could possibly

indicate that PVIs were possibly better at utilizing spatial audio cues for orientation than the blindfolded participants.

For Task 2: Significant differences were observed between G2 ($p < 0.001$), G3 ($p < 0.001$), and G4 ($p < 0.001$) when compared to G1. This is expected since G1 had all the sensory cues including vision. No significant differences were observed between the completion times for task 3 between G2 and G3 ($p = 0.12$), G2 and G4 ($p = 0.23$), and G3 and G4 ($p = 0.51$). It appears that the effect of distance estimation overpowers that of direction localization (task 1). Given that distance estimation is known to be difficult through auditory feedback, the lack of differences is expected. Having said that, we do observe a higher mean, median and inter-quartile range for G3 ($\mu = 26.38$ sec, M = 20.99 sec) as compared to G2 ($\mu = 24.28$ sec, M = 17.28 sec) and G4 ($\mu = 21.38$ sec, M = 17.93 sec). This suggests that even with the difficulty of distance estimation, the lack of spatial audio for G3 is what likely resulted in wider spread of the distribution.

## 5.2 Analysis of Action Phases

We compare the time taken in each of the four phases by each participant (See Fig. 5). We do this comparison across the two tasks for each user group. Similar to the previous analysis, we first tested the data for normality using Shapiro-Wilk test and found out that the data is not from a normal distribution. We further conducted hypothesis testing using Kruskal-Wallis non-parametric test ($\alpha = 0.05$). We performed pair-wise comparison of the time taken in each phase by the participants in each group across the two tasks. Similar to the previous analysis, we considered group 1 as the gold standard.

For Task 1: It is important to note that no significant differences were found between G2 and G4 for mean times in initial reaction time ($p = 0.1$), approach phase ($p = 0.23$), stabilization phase ($p = 0.94$), and slider pulling phase ($p = 0.22$) indicating that the perceptual experiences of the sighted blindfolded and the PVIs were roughly similar in the orientation task. Significant differences were observed in the mean initial reaction time for each trial between G2 and G3 ($p < 0.001$), and G3 and G4 ($p < 0.001$). The shorter reaction times for G3 compared to G2 and G4 could be explained by the fact that without the spatial audio cues G3 participants had to move to even find out the relative position of the target to their current orientation, whereas the other two groups had to spend some time extracting this information from the spatial audio cue before starting their movement. Mean time for stabilization phase was significantly more for G3 and G4 ($p = 0.04$) with much higher standard deviation

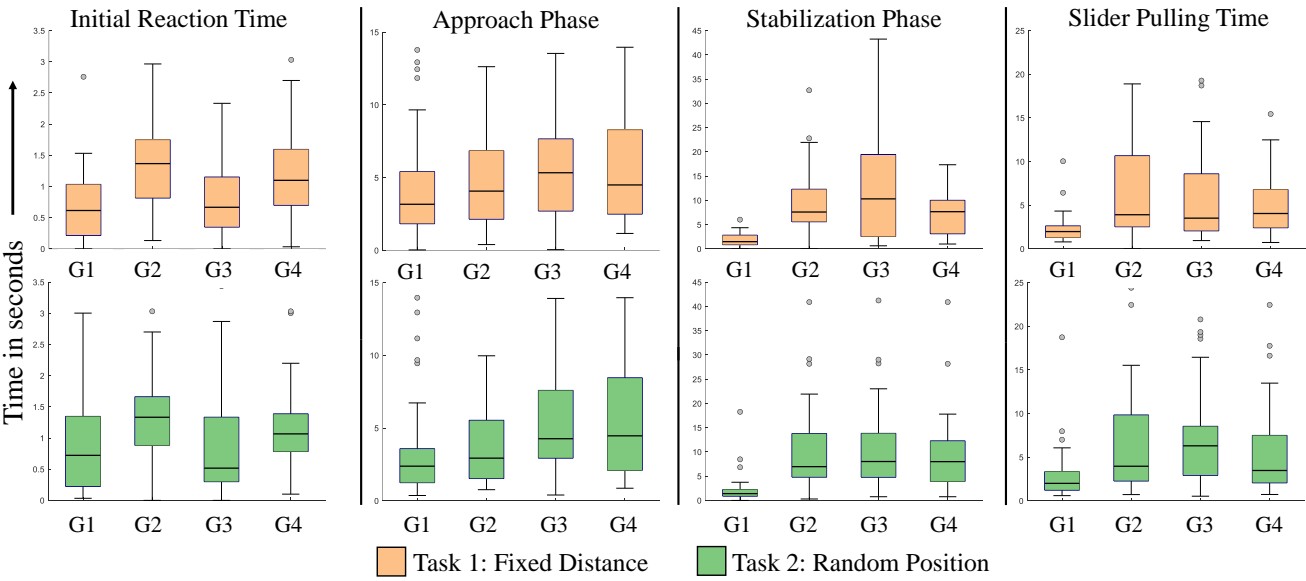

Figure 5: Time taken for each phase for the three tasks across the four groups

indicating the utility of spatial audio cues.

For Task 2: The relative differences between various groups and their statistical significance were roughly similar to that of Task 1. In general, all the time durations became lesser than for Task 1 indicating the learning effect on the participants since Task 2 was the last Task to be performed.

## 6 DESIGN IMPLICATIONS

### 6.1 Hardware Challenges

A key methodological issue we faced in our study was the difference between the protocols for the sighted users and PVIs. While the study with sighted individuals was conducted with a single smartphone in person, the study with PVIs was conducted remotely with each PVI user using their personal smartphone. The lack of consistency in the smartphone models resulted in two critical hardware-specific issues because of which two of the enrolled participants were unable to get a single hit in 10 trials. As a result, we terminated the study in order to prevent discomfort and feeling of inadequacy by conveying to them that there must be hardware issues.

Upon looking closely at the smartphone IMU data, we found that these users primarily faced issues due to abnormally high noise resulting in complete loss of control over the game. Further analysis showed that the vibrotactile feedback generation was very compute intensive. Given that the study was being conducted while on call with the researcher, the Talkback and vibrotactile feedback being a load on the CPU, we speculate that there is a likelihood of latencies that depend on the phone in the audio lock feedback. Finally, we also faced difficulties with different aspects of the smartphone such as battery drainage, audio-lock cue being delayed causing them to miss, and difficulty with Talkback gestures being recognized. An earlier work on Talkback with PVI users also indicated the issues with touch screen sensitivity and gesture recognition being non-uniform across phones [44]. Thus hardware and OS versions should be carefully considered by researchers working with audio and tactile interfaces on smartphones for PVIs.

### 6.2 Future Challenges with Distance Estimation

Even though the study mainly focused on orientation, we also collected some preliminary data of the ranging aspect of the task which was facilitated by the vibrotactile feedback and the lock sound. We

recorded the number of times the participants were able to successfully hit the balloon. Since a successful hit requires both orientation and distance estimate to be correct, the number of hits gave us a few insights on the ranging aspect of the task. We observed lower number of hits for task 2 which combined both orientation and ranging as compared to task 1 across all four user groups. This may be due to the additional ranging task that users had to do for task 2. We observed that the vibrotactile feedback for distance estimation was not effective and it is not a trivial problem. In retrospect this is expected because touch is a near sense and has no bearing on far distance estimation from an evolution standpoint. Hence what we are attempting to do is sensory substitution [82] where with repeated usage the amplitude of the vibration could be mapped to the distance of the target. Our blindfolded subjects and PVIs relied on the lock sound for determining the range rather than the amplitude of the vibrations. Our sighted non-blindfolded participants used the color cue to determine when the slider has been pulled to the right amount. We also observed a monotonous decrease in the number of hits when going from G1 to G4. Participants of G2 performed better than that of G3 in terms of no. of hits which further highlights the effectiveness of spatial audio in orientation tasks. PVIs had much worse hit percentage than the other groups, even though based on the analysis of their phase-wise performance, it is clear that they were able to orient the phone with similar processes as the other groups. We believe that the hardware challenges as discussed in the previous section were the main reason for lower hits by PVIs. We believe that there is yet a richer set of questions regarding tactile cues for PVIs that warrants an isolated study to determine the effectiveness of the tactile feedback for range estimation.

### 6.3 Potential interaction guidelines for PVI

Another avenue for future research is the study of the ergonomics and affordances of a smartphone in such accessible AR applications. For instance, some of our participants held the phone in one hand and used the other hand to pull the slider while there were a few who used one hand to do both. Some participants used the device while standing while some others performed the trials while sitting down. Since the studies with PVIs were done remotely on a single device, we did not have the opportunity to study these aspects in detail. We asked a few questions about their method of use, but a controlled study of cue perceptions in different configurations is

important to arrive at design and interaction guidelines for diverse end applications.

## 7 CONCLUSION

In this paper, we sought to develop a fundamental understanding of two specific spatial tasks, namely, orientation and ranging for persons with visual impairment. The motivation to do so stemmed from the fact that affordable mobile AR needs a wide variety of guidelines for system, interaction, and feedback mechanism design. While we experimented with both spatial audio and vibrotactile feedback, we observed that spatial audio cues provide sufficient help in orientation tasks. The next stage of the work is to extend the interaction to 360 degrees around the user instead of just on the azimuthal plane while continuing our effort to provide the means for precise distance estimation. Systematic research is required to explore new sensory cues which will make the distance estimation easier. There is a need to explore and study the role of vibrotactile feedback in tasks which require manipulating objects close to the body. Having said that, this research is still only a glimpse of the rich research that is yet to be done in the domain of accessibility in AR systems.

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
