# OpenReview forum: "Pointing Tasks using Spatial Audio on Smartphones for People with Vision Impairments"
_graphicsinterface.org/Graphics_Interface/2022/Conference — Submitted to GI 2022_

### Official Review · Reviewer_FBm6 · 2022-01-06
**Paper 13 review**

**Rating:** 5
**Confidence:** 3

**Review:**

The paper describes an experiment to investigate spatial audio feedback using smartphones for people with visual impairments. The research discusses how it can support direction localization in pointing tasks by using a mobile game with the metaphor of a bow-and-arrow task. The paper discusses the efficacy of spatial audio and general behaviour in three distinct groups of visual impairment.
The authors have put the effort into describing the background and reasoning behind their experiment design and decisions. Overall, the paper discussed the differences and similarities between groups in depth under the well-constructed experiment design.
Below are my comments:
The design restricts users to hold phones parallel to the ground, rather than perpendicular. This will exclude the possibilities of apps using cameras for AR as many smartphones have a similar form factor with the cameras looking at the directions perpendicular to the display/surface. While the authors claimed that their target is to make accessible AR applications, the assumption would exclude many existing AR apps in which were designed for users without visual impairment. It would be interesting to see the study around perpendicular setup as well for future works.
While focusing on spatial audio cues is interesting, using vibrotactile feedback could be improved using different patterns of vibrations. Things could indicate different colour/target types or so because the colour seems to be only available to non-visual impaired users.
It would be interesting to see an analysis of the rate of change between groups for the angular error in the approach phase.

---

### Official Review · Reviewer_utRR · 2022-01-14
**concerned about the validity of the results given the confounds of how the experiment was conducted**

**Rating:** 3
**Confidence:** 4

**Review:**

This paper presents an experiment that seeks to understand the impact of spatial audio on smartphones for AR-style pointing tasks across sighted individuals and people with a visual impairment (PVIs).

There is a lot that I really like about this paper. Overall, it is quite well written, and the motivation is particularly strong. The research question being addressed is important, with the potential for high contribution to the community.

The reason the I cannot recommend any level of acceptance for this paper is that I don’t believe the results are valid due to the confounds in how the study was conducted. The authors are entirely forthcoming about the substantial differences between how sighted participants were run through the experiment versus how it was done for PVIs. The problem is that the paper puts forward specific results (in the abstract and Section 1.3) that compare performance for these groups, and I just don’t see any way they can be seen as valid. It is not good practice for results to be published where their validity is entirely in question. Some people in the academic and industrial communities could easily rely on these results without actually reading the paper to realize that they need to be used, at absolutely best, as suggestive.

I definitely sympathize with the challenges faced by the authors during the pandemic to conduct this research. My best advice would be to re-run the entire study when physical distancing requirements are relaxed. At a minimum, the PVI group needs to be re-run.

Here are some additional comments:
-	Writing issue: framing in 1.1 is a bit odd with “the question we seek to address is:…” then later followed by “we begin by asking the following questions…”
-	High-level: if the main goal is to devise techniques/support to assist PVIs, why is the main comparison of performance against sighted people? Wouldn’t it be more valuable to compare different forms of support for PVIs to uncover which provide the better support?
-	Writing: in 1.2 it is written that “There are few works that discuss VR accessibility for PVIs.” But the paragraph then goes on to list quite a good number of references.
-	Writing: quite a lot of repetition between 2.2 Perceptual Cues and 3.1 Game Design, and in 2.2 I kept wondering what the spatial audio was that was being used (then clarified in 3.1).
-	4.1 mentions 12 participants had complete loss of vision, yet later on for Group 4 says “All the Blind and Visually Impaired”, which seems to suggest that some participants did not have complete loss of vision.
-	Throughout the paper it is mentioned that sighted participants are blindfolded. I was taken aback in 4.1 where it mentions that participants in G2 and G3 just played the game with their eyes closed. Beyond the confusion in the writing, having participants only close their eyes seems questionable to me (unless it can be supported by the literature), and seems like it could be an internal validity issue.
-	Higher precision of the phases is needed. For example, for the Initial Reaction Phase “This phase starts when the trial starts, i.e., the participant starts to hear the beeping sound.” What objectively marks the beginning of a trial? The study design cannot determine when a participant hears something, only when a cue is given. Another example is from the Stabilization Phase: “… ends as soon as the phone movement stops or becomes relatively small.” How is relatively small quantified/assessed?
-	I didn’t understand Fig 3(b), in particular why the 4 representative trial lines all have different angular error to start. Are these 4 different representative trials, meaning a different trial is shown for each person in the 4 groups? We should see the same trial that was done by 4 different participants, one from each group. The angular error be the same to start, as all participants should start in the same position relative to the target. Or perhaps I am missing something.
-	Isn’t median a better measure for analysis on central tendency than mean?
-	Were the pair-wise comparisons corrected for type 1 error?
-	High-level: is the goal of the research to understand initial usability of spatial audio OR more expert use of it? The duration of the study in terms of total trials, including the number of practice trials before performance was measured, seems very low to me. I would be interested to see performance curves that show trials over a longer period of time (with adequate breaks of course). While initial usability is interesting, given the motivation for the research, it seems that the authors are more interested in skilled behaviour which I do not believe is captured in this study.


I hope these comments are helpful and wish the authors success with their work.

---

### Official Review · Reviewer_iziF · 2022-01-17
**B/LV orientation and distance judgement in the horizontal plane using spatialized audio**

**Rating:** 4
**Confidence:** 4

**Review:**

This paper reports on an assessment of using spatialized audio, vibrotactile and visual cues on the assessment of target orientation to and distance from user in the horizontal plane for sighted, blind-folded sighted and people who are blind/low vision (B/LV). The main findings are that there are significant differences between sighted users, and blind-folded and B/LV users in time to locate and acquire targets, and that spatialized audio can be used to successfully acquire targets by sighted and B/LV users. While this research is interesting and potentially useful in understanding the importance of spatialized audio in VR for people who are BLV, the comparison with sighted users (fully sighted or blind-folded) is less so. Blind-folding sighted users is not a good replacement for B/LV users and should not be suggested as an alternative to evaluating designs with B/LV users for many different reasons including mental models of visual artefacts, differing abilities to process and understand sound environments, etc. In addition, it would have been a useful addition to understand what the users thought of the tasks and the usefulness of the audio..
Specifically, concerns are:
1.	Why are blindfolded sighted users a factor in this when this would not be necessary in AR? I don’t understand the motivation for, or important of using blindfolded sighted users?
2.	The notion of producing guidelines for sighted hci designers based on this study seems over claimed and not the focus of this work. There are no hci designers in this study.
3.	How was “normal” hearing capability measured?
4.	A declaration of approval from an research ethics board is missing from this paper.
5.	Why would median and mean be used when mean/SD is sufficient and appropriate? Standard deviation must be reported with mean, and interquartile range with median. The Kruskal Wallace H values should be reported with the related p-values. I do not understand why the authors provide box plots and used mean and SD?
6.	When there is no significance difference, it is does not mean things are similar, only not different because there could be a variety of reasons (e.g., insufficient participant numbers is the most common reason). To show similarity between 2 groups requires considerable more work with respect to power, normality, etc. Any claims about similarities should be removed.

Technical
1.	Do not begin a paper with a figure.
2.	“Has been” is repeated 2x in the same sentence in section 1.1 paragraph 2.
3.	Do not use “on the other hand” without “on the one hand” appearing before it.
4.	Last paragraph of section 1.1 has an incomplete sentence.
5.	Missing space after 1st sentence of last paragraph of prior work section.
6.	Use full words and not contractions in academic writing.
7.	Use a single verb tense in each paragraph. Do not mix verb tenses within a paragraph – this occurs frequently in this paper.
8.	There is considerable repetition in this paper (e.g., game design and interactive context).
9.	Begin sentences with words not numbers.
10.	IMU data = ?
11.	There are numerous other grammar errors and awkward sentences. A thorough edit is required.

---

### Decision · Program_Chairs · 2022-01-18

Reject